# Photocatalytic Removal of Harmful Algae in Natural Waters by Ag/AgCl@ZIF-8 Coating under Sunlight

**Gongduan Fan [1,2,\*]** **, Zhong Chen [1], Bo Wang [3], Shimin Wu [4,\*], Jing Luo [1], Xiaomei Zheng [1], Jiajun Zhan [1], Yifan You [1] and Zhi Zhang [5,\*]**

[1] College of Civil Engineering, Fuzhou University, Fuzhou 350116, China
[2] State Key Laboratory of Photocatalysis on Energy and Environment, Fuzhou University, Fuzhou 350002, China
[3] School of Urban Planning and Design, Shenzhen Graduate School, Peking University, Shenzhen 518055, China
[4] IER Environmental Protection Engineering Technology Co., Ltd., Shenzhen 518071, China
[5] College of Environment and Ecology, Chongqing University, Chongqing 400044, China
\* Correspondence: fgdfz@fzu.edu.cn (G.F.); wushimin@gmail.com (S.W.); zhangzhicq@126.com (Z.Z.)

**Abstract:** In order to control the cyanobacterial blooms in eutrophic water, an Ag/AgCl@ZIF-8 floating coating was prepared by a dip-coating method with a sponge, innovatively employed as a carrier for the removal of algae in natural water samples. The as-prepared photocatalyst was characterized by X-ray diffraction (XRD) and field emission scanning electron microscopy (SEM). The effects of this Ag/AgCl@ZIF-8 coating on algal removal and phytoplankton community structure in natural water samples with cyanobacterial blooms were investigated under sunlight. Results showed that Ag/AgCl@ZIF-8 distributed uniformly on the surface of the coating with good stability and algae removal efficiency in water bodies. After 6 h of exposure under sunlight, the chlorophyll *a* in the natural water samples was degraded by 99.9%, the densities of *Microcystis aeruginosa* were reduced by 92.6% and the densities and biomass of the other algae decreased by about 80%. Meanwhile, the content of colored dissolved organic matter (CDOM) in the samples was decreased, effectively controlling the cyanobacterial blooms. It was found that $O_2^{\bullet-}$ played the main role in the photocatalytic inactivation. In conclusion, the Ag/AgCl@ZIF-8 coating has a promising application potential for the removal of harmful cyanobacteria, and provides a new idea for the control of cyanobacterial blooms in water bodies.

**Keywords:** Ag/AgCl@ZIF-8; coating; harmful algae; photocatalysis; sunlight

## 1. Introduction

With the acceleration of urbanization and industrialization, eutrophication and climate change are causing a global increase in the frequency and severity of harmful cyanobacterial blooms [1]. It has been reported that nearly half of the lakes in Europe, Asia and America are eutrophicated, and a quarter of them have recorded outbreaks of cyanobacterial blooms [2], which has a long-term negative impact upon drinking water quality, fishery and tourism [3,4]. Up to now, numerous techniques have been proposed to control cyanobacterial blooms, including physical [5,6], chemical [7,8] and biological methods [9,10]. In recent years, the application of photocatalysis has been studied in controlling cyanobacterial blooms. Photocatalysis is the acceleration of a chemical reaction stimulated by light with the aid of a photocatalyst, which is usually a semiconductor [11]. The combination of semiconductors and the way of combining them are also important for the catalyst response. Moreover, the dispersion grade of semiconductors and the catalyst or support presents an influence on reaction efficiency and band gap [12].

Since it was found that the TiO$_2$ electrode can decompose water under light conditions [13], photocatalytic oxidation technology has attracted increasing attention from researchers worldwide. With the rapid development of this technology, it shows great potential for future application in the field of water treatment.

At present, nanomaterial is one of the most popular photocatalysts for its high activity, good chemical stability, being non-toxic, easy to obtain, at low cost, an abundant raw material and possessing strong adsorption ability [14], and has been widely used in various photocatalytic oxidation reactions, which strongly promote research in pollution control. The Nano-photocatalyst can produce reactive oxygen species (ROSs) under irradiation, and the generation of ROSs can cause irresistible damage to microorganisms and various chemical pollutants [15,16]. It is an effective and low-cost new control method for cyanobacterial blooms.

Metal-organic frameworks (MOFs) are a kind of porous crystals with periodic multidimensional network structure, formed by coordination of metal ions and organic ligands, and have been widely used in many fields, such as separation [17], adsorption [18], catalysis [19], gas storage [20], sensor [21], and battery [22], based on their large specific surface area, high porosity, good thermal stability and easily-controlled physicochemical properties [23,24]. MOF materials have also achieved remarkable results in the environmental field [25–27], being considered as one of the most effective methods to solve current environmental problems [28]. Therefore, MOFs have broad prospects in preventing, controlling and mitigating harmful algal blooms in waters.

However, the research on using MOF nanomaterials to control algae is still in its infancy, and several problems exist in practical use, as follows: (1) The existing catalysts are mainly in powder form, which is difficult to be recycled, and may cause secondary pollution that adversely affects aquatic organisms [29]; (2) Most of the catalysts are suspended in water, which is suitable for homogeneous pollutants in water, while algae mainly grow upon the surface of the water [30]; (3) Pure MOFs need to improve the catalytic performance by modification for their limited catalytic capacity [31]; (4) There are more complex substrate background and various species of algae in natural water, which will affect the photocatalytic removal of *Microcystis aeruginosa* (*M. aeruginosa*).

It has been widely reported that ZIF-8, constructed from Zn$^{2+}$ and imidazolate organic ligands, exhibits higher thermal and chemical stability compared to other MOFs [32]. Ag and Ag compounds are often used as a modified material for the synthesis of catalysts or heterostructures [33], due to their electron trapping ability and localized surface plasmon resonance (LSPR) effects [34]. In our previous study [35], MOFs modified by Ag/AgCl have a good degradation effect on pharmaceutical and personal care products (PPCPs). Therefore, Ag/AgCl was selected as the modifier of ZIF-8 to study its effect and the mechanism of algae removal. In this study, ZIF-8 was modified by Ag/AgCl to synthesize Ag/AgCl@ZIF-8, improving the light absorption properties and catalytic activity, and this Ag/AgCl@ZIF-8 coating was prepared by a dip-coating method using a sponge treated with sodium dodecyl benzene sulfonate (SDS) as a carrier. As a common load material, the sponge is easy to promote in practical applications for its advantages of low cost and easy access. Additionally, the sponge is a porous material that can float upon the surface of the water and adsorbs large numbers of algae cells, which is beneficial for the material to absorb sunlight and to have contact with algae on the surface of the water body. Floating and coating material can not only improve the efficiency of photocatalytic algae removal, but is also beneficial to recovery and reuse to avoid secondary pollution.

In this paper, the natural cyanobacterial bloom samples were taken as the research object to explore the catalysis potential of Ag/AgCl@ZIF-8 coating to solve the natural bloom problem under sunlight. The effects of the Ag/AgCl@ZIF-8 coating on *M. aeruginosa* and other phytoplankton in natural water samples were studied by detecting the density and biomass of phytoplankton, the content of chlorophyll (Chl) *a*, the changes of community structure, and its diversity index. Both the effect and mechanism of algal removal by the new nano-coating material in natural waters were further investigated, thus providing a theoretical foundation for the future application of Ag/AgCl@ZIF-8 coating in the natural water bodies.

## 2. Results and Discussion

### *2.1. Characterization of Photocatalysts*

The crystallographic structure of the prepared Ag/AgCl@ZIF-8 powder and coating were examined by XRD. The XRD patterns of Ag/AgCl@ZIF-8 powder and coating are illustrated in Figure S4. Ag/AgCl@ZIF-8 exhibits diffraction peaks at approximately 2θ = 27.89°, 32.30°, 46.30°, 54.88°, 57.52° and 76.78°, which can be ascribed to the (111), (200), (220), (311), (222) and (420) reflections of AgCl [36]. Except for the distinct diffraction peaks of AgCl, the characteristic peaks at 2θ = 7.39°, 10.42°, 12.78° and 18.08° can be ascribed to the (011), (002), (112) and (222) reflections of ZIF-8 [27]. All peaks are sharp and clear, suggesting that Ag/AgCl@ZIF-8 was successfully synthesized. Meanwhile, except that the peak intensities of Ag/AgCl@ZIF-8 coating were weaker than that of powder, all characteristic peaks of the coating were well consistent with that of powder, indicating that the structure of Ag/AgCl@ZIF-8 does not change after modified with SDS.

As shown in Figure 1a,b, the blank sponge was light yellow, and the appearance of the sponge was purple-black after being loaded with Ag/Agcl@ZIF-8, indicating that Ag/Agcl@ZIF-8 was successfully loaded onto the sponge. It can be seen from the SEM images (Figure 1c,e) that the blank sponge has a three-dimensional network structure with a smooth surface and no impurities. The microstructure of this sponge reveals its macroporosity. These pores may help in the transportation of reactant/product during the reaction. Also, its porous surface allows an immobilization of more catalysts for more surface reaction sites. The surface of the sponge became rough after being loaded with Ag/Agcl@ZIF-8 (Figure 1d,f), and the photocatalysts almost covered the skeleton of the sponge completely, and distributed evenly without excessive accumulation. This indicated that the dip-coating method could effectively and uniformly load Ag/AgCl@ZIF-8 onto the floating carrier. According to our previous studies [35], the zeta potential of Ag/AgCl@ZIF-8 exhibits a positive surface charge in a neutral environment. SDS is a commonly-used anionic surfactant that can attract Ag/AgCl@ZIF-8 to the sponge treated with SDS by electrostatic effect. Meanwhile, 2-methylimidazole of ZIF-8 contains an imidazole ring, which can also be attracted to the surface of the sponge by π-π interaction with the benzene ring of SDS. The loading of the catalyst did not change the pore structure of the sponge, and the carrier still maintained the original physical properties, meeting the test requirements.

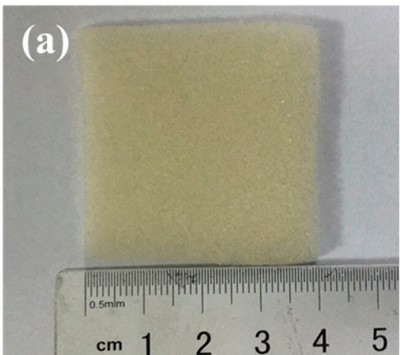
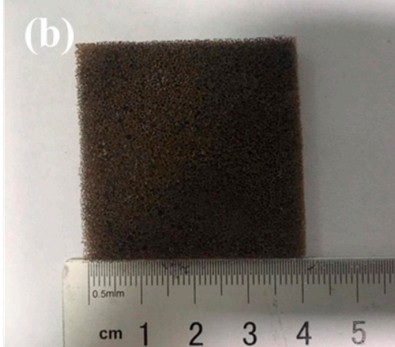

**Figure 1.** *Cont.*

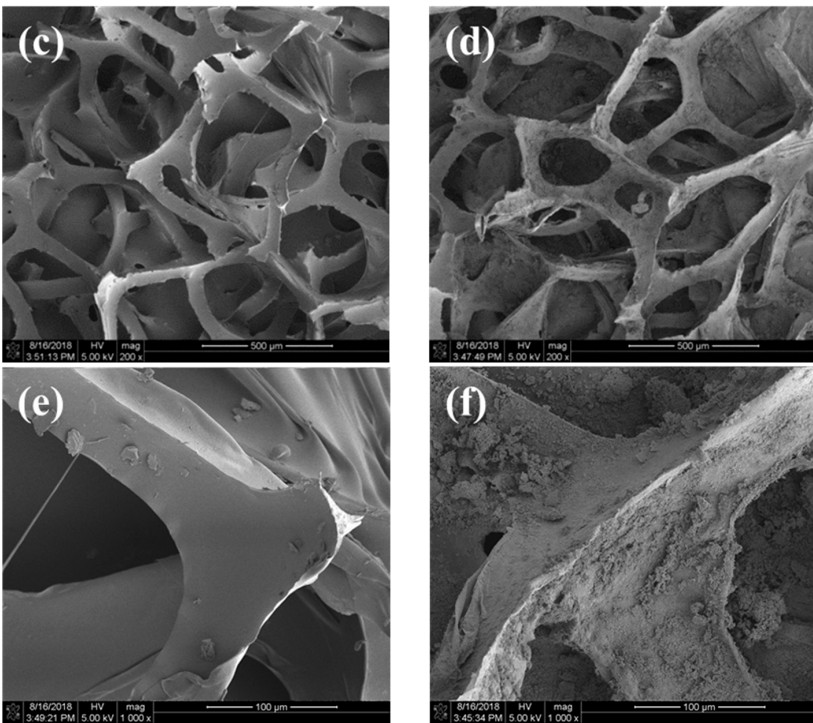

**Figure 1.** Photos of sponge (**a**) without photocatalyst and (**b**) with Ag/AgCl@ZIF-8 coating, scanning electron microscopy (SEM) images of sponge (**c**), (**e**) without photocatalyst and (**d**), (**f**) with Ag/AgCl@ZIF-8 coating.

## 2.2. Effect of Coating Treatment on Algae in Environmental Water Samples

### 2.2.1. Morphological Changes in Algal Cells

The change of cell morphology and color of samples are shown in Figure 2. It can be seen that the natural water sample was yellow with high chrominance. After 6 h of photocatalytic reaction, the color of the algal solution in the control group did not change, while the color of the algal solution in the experimental group was obviously faded, and the solution became transparent and clear. Before the experiment, the cell morphology of *M. aeruginosa* was regular, and the cells were stained uniformly by Lugol's iodine solution. However, the cells of *M. aeruginosa* were agglomerated and became greenish, and could not be stained by Lugol's iodine solution after the reaction. Furthermore, it can be seen from Figure 2b,c that the boundaries between cells and the outline of the cells became blurred, which implied the cell structure was dissolved. These results suggested that the Ag/AgCl@ZIF-8 coating can cause irreversible damage to algae cells in natural water samples under sunlight.

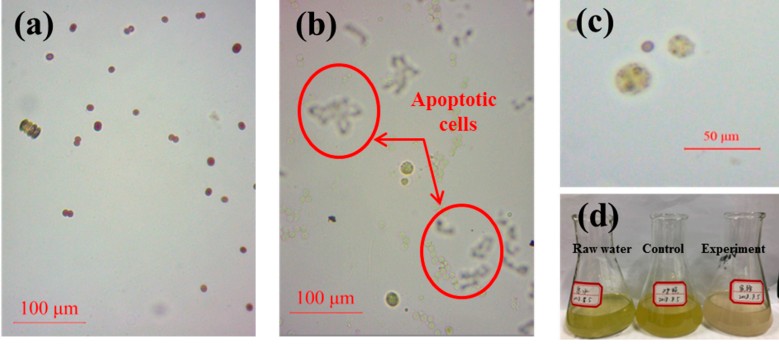

**Figure 2.** Cell morphology (**a**) before and (**b**) after the experiment of *M. aeruginosa*, (**c**) *M. aeruginosa* and (**d**) changes of color of algal solution.

### 2.2.2. The Density and Biomass of Phytoplankton

Chl *a* acted as an essential pigment in the photosynthetic system of algal cells. As can be seen from Figure 3a, the content of Chl *a* was reduced significantly by the Ag/AgCl@ZIF-8 coating. After the reaction for 1 h, more than 20% of Chl *a* was degraded, and the Chl *a* content in the algal solution was rapidly reduced after 4 h, indicating that the algal cell defense system collapsed at this time, and the oxidative stress of the external environment caused a large number of the algal cells to die. As the reaction continued, Chl *a* was continuously degraded, and the content of Chl *a* in the water sample was almost reduced to zero after the reaction. In contrast, the Chl *a* content in the control group showed no apparent changes during the reaction.

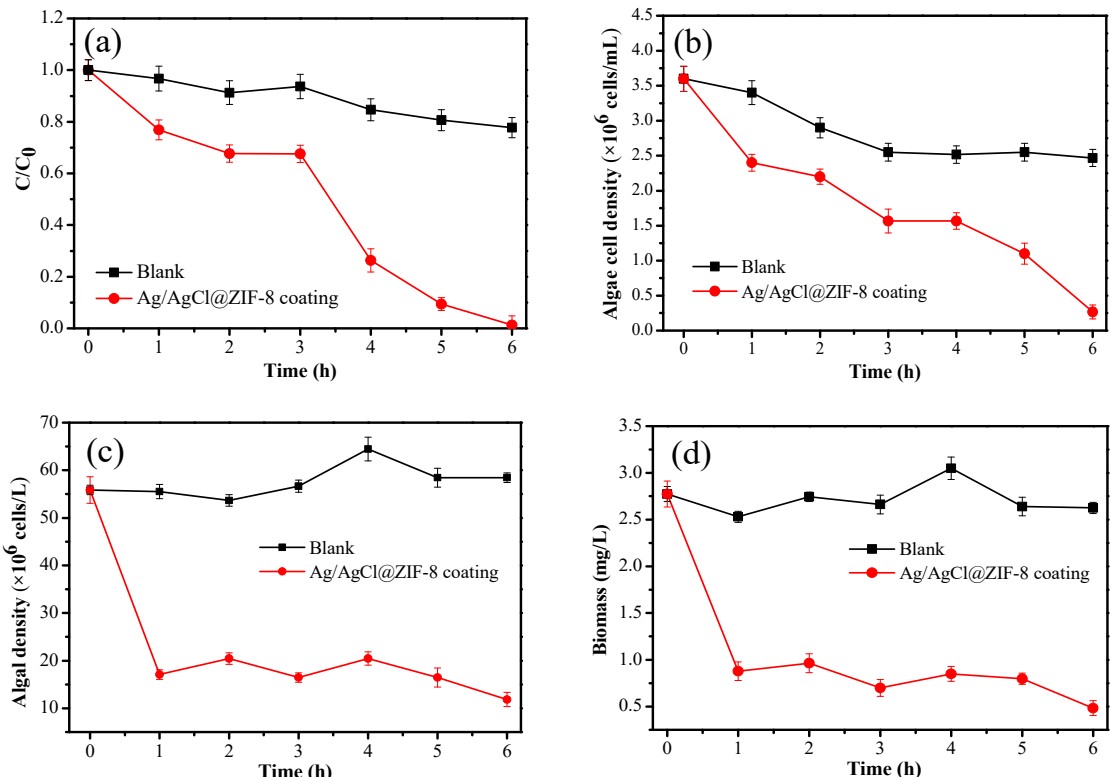

**Figure 3.** Changes of (**a**) relative chlorophyll (Chl) *a* content, (**b**) density of *M. aeruginosa*, (**c**) density and (**d**) biomass of other phytoplankton.

The changes in algal density in the solution can directly reflect the inactivation of *M. aeruginosa*. As can be seen from Figure 3b, the initial density of *M. aeruginosa* in the natural water sample was $3.6 \times 10^6$ cells/mL. After 1 h, the algal density of the experimental group decreased to $2.4 \times 10^6$ cells/mL, which decreased by almost a third. After 3 h, more than half of the *M. aeruginosa* were inactivated, and after the reaction, the density of *M. aeruginosa* was only $2.7 \times 10^5$ cells/mL, which was 7.4% of the initial density. The density of algae in the experimental group decreased obviously during the whole reaction process. However, a reduction of the density of algae which occurred in the first 3 h can be ascribed to the increase in water temperature. The density of algae in the control group remained stable afterwards.

The density and biomass of other phytoplankton changes in samples during the photocatalytic treatment are shown in Figure 3c,d. Except for *M. aeruginosa*, the density of phytoplankton in the solution was $55.9 \times 10^6$ cells/L before the experiment. After 1 h of the experiment, the density of phytoplankton in the solution decreased rapidly to $17.1 \times 10^6$ cells/L, which was 30.6% of the initial amount. Then, the density decreased slowly to $11.8 \times 10^6$ cells/L at the end of the experiment. However, the density of phytoplankton in the control solution did not change significantly during the

photocatalytic process, indicating that the growth of phytoplankton except for *M. aeruginosa* was not affected by sunlight in a short time. The changing trend of biomass corresponded to that of density. Before the start of the reaction, the biomass of phytoplankton except for *M. aeruginosa* in the solution was 2.77 mg/L, and after 6 h, the biomass was significantly reduced to 0.48 mg/L, which was 17.3% of the initial value. The biomass of the control group did not change significantly during the reaction. This indicated that the Ag/AgCl@ZIF-8 coating could effectively reduce the density and biomass of phytoplankton in natural water samples under sunlight. The coating performed a significant effect upon algae removal with low dose and high removal rate, compared with other photocatalysts reported in the literature. A comparison with other photocatalyst has been shown in Table S2.

### 2.2.3. Community Structure of Phytoplankton

Changes of phytoplankton biomass and community structure over time during photocatalysis were shown in Figure 4. It can be seen that the total biomass in the initial water sample was about 2.7 mg/L, and there was no significant change of the phytoplankton community structure in the photocatalytic reaction process in the control group. After 1 h of reaction, the total biomass in the experimental group reduced from 2.7 mg/L to 0.8 mg/L; the biomass of *Chlorophyta* reduced from 1.67 mg/L to 0.63 mg/L, that of *Bacillariophyta* reduced from 0.66 mg/L to 0.09 mg/L, that of *Cyanophyta* reduced from 0.20 mg/L to 0.06 mg/L, and continued to reduce over time. After 6 h, the total biomass in the water was less than 0.5 mg/L, which was 80% less than the initial value.

It can be seen from Figure 4b that there were no significant changes in the phytoplankton species in the control group, while that in the experimental group decreased significantly. After 6 h of photocatalytic oxidation, the species of *Chlorophyta* was reduced by half, and *Euglenophyta*, *Chlorophyta* and *Cryptophyta* were not detected. The total species of *Chlorophyta* was reduced by more than half from 26 to 11. In the photocatalytic process, the species of *Chlorophyta*, *Bacillariophyta* and *Chlorophyta* in the control group and the experimental group were always the main species, and the number of the three phyla reached more than 80% of the total number. In the control group, there was no significant change in phytoplankton community structure during the reaction process, with *Chlorophyta* accounting for about 50%, *Bacillariophyta* accounting for about 22% and *Cyanophyta* accounting for about 18%. Thus, the community structure of phytoplankton in the experimental group showed dynamic fluctuation, with the proportion of *Cyanobacteria* being decreased to only 10%. The results show that the Ag/AgCl@ZIF-8 coating could alleviate the problem of algal bloom by reducing the proportion of *Cyanobacteria* in phytoplankton under sunlight.

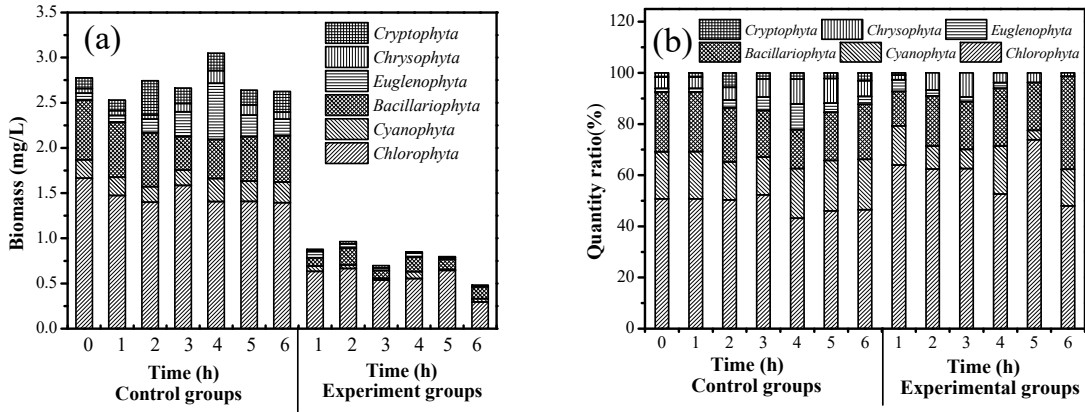

**Figure 4.** Changes of algae (**a**) biomass, and (**b**) community structure.

### 2.2.4. Diversity Index of Phytoplankton

The dynamic changes in the diversity index were shown in Figure S5. The Shannon-Werner index, richness index and uniformity index of the natural water samples were 2.272, 4.0717 and 0.7101,

respectively. During the experiment, the Shannon-Werner index of the control group was always above 2, indicating that the species diversity of water samples was always at a high level. The range of the Shannon-Werner index, richness index and uniformity index were 2.272–1.468, 4.0717–2.3021 and 0.7101–0.5723, respectively, which were always lower than those of the control group. Besides, all of the three indices show an attenuation trend, which indicates that Ag/AgCl@ZIF-8 coating could reduce the diversity of phytoplankton in water samples under sunlight.

### 2.2.5. CDOM

CDOM plays a vital role in DOM, which can absorb ultraviolet light and blue light and be colored [37]. The high concentration of CDOM in the lake will significantly increase the absorption of light by the water bodies, whereas it reduces the reflectance of water bodies to light, which is an important reason for the occurrence of black bloom [38]. CDOM is mainly composed of proteinoid and Humic-like substances. Generally, it comes from the input of exogenous humic acid and the endogenous degradation of aquatic plants [39]. Moreover, it was found that changes in CDOM content can significantly change the community structure of microorganisms in water [40]. Therefore, it is of considerable significance to pay attention to the content change of CDOM in natural water to evaluate the effect of the Ag/AgCl@ZIF-8 coating on algal removal in natural water.

Three-dimensional fluorescence scanning was performed on the algal solution in the reaction process, and the results were shown in Figure 5. Three peaks were observed in the fluorescence spectrum with $\lambda_{ex}/\lambda_{em}$ = 285 nm/325 nm, 225 nm/325 nm, and 250 nm/450 nm, which were ascribed to be tryptophan-like, tyrosine-like and fulvic-like, respectively. The original sample contained only two fluorescence peaks of tryptophan-like and tyrosine-like, which represented the biochemical organics with rich activities. With the increase of photocatalytic treatment time, the content of tryptophan-like and tyrosine-like gradually decreased, indicating that the biological activity in the water sample decreased. At 4 h after the reaction, fulvic-like was found in the solution, which was attributed to tryptophan-like and tyrosine-like being degraded and transformed into fulvic-like. In the control group, tryptophan-like and tyrosine-like increased first and then decreased, which was mainly because the light in the first 2 h stimulated some intracellular enzymes that promoted the proliferation of algal cells. Then, algal cells adapted to the stimulation of sunlight and returned to the normal growth state. These experimental results demonstrate that the Ag/AgCl@ZIF-8 coating could degrade algal metabolites and effectively avoid secondary pollution caused by algal cell rupture when applied to remove *M. aeruginosa* in natural waters.

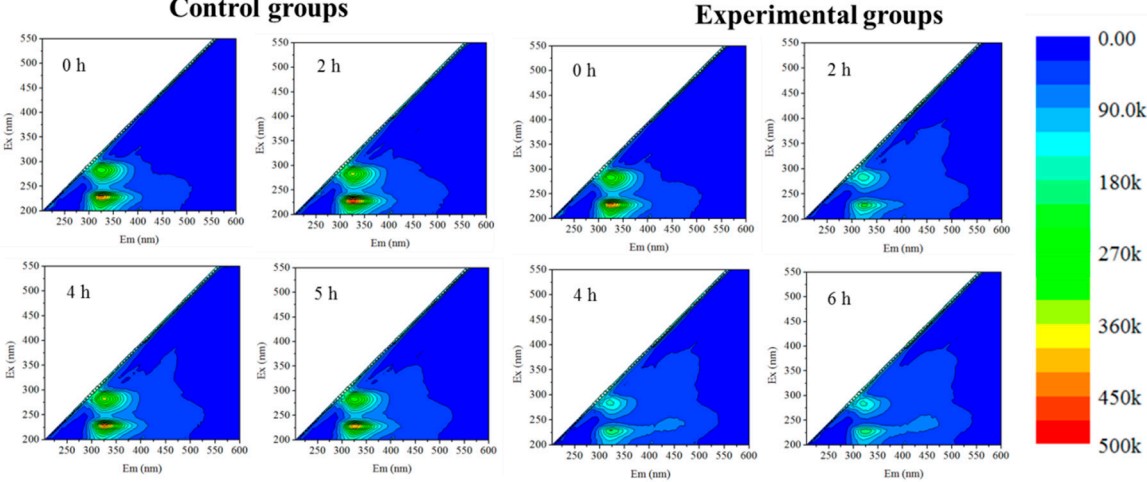

**Figure 5.** Three-dimensional fluorescence spectrum of colored dissolved organic matter (CDOM) during the photocatalytic removal of algae.

### 2.3. Stability of Coating

Four recycling runs of photocatalytic degradation experiments were conducted to illuminate the stability of our Ag/AgCl@ZIF-8 coating (Figure S6). A slight decrease from 99.9% to 94.0% was noticed for the photo-degradation of Chl *a* after the fourth cycle. After the reaction, no obvious Ag/AgCl@ZIF-8 solid was observed at the bottom of the beaker, and the appearance of Ag/AgCl@ZIF-8 coating showed no significant changes, indicating that the coating prepared by the dip-coating method had good adhesion and recyclability. The reduction of photocatalytic activity of the catalyst was attributed to the loss in the recycling and cleaning process.

### 2.4. Mechanism of Photocatalytic Algal Removal by Ag/AgCl@ZIF-8 Coating

The result of quenching experiments is shown in Figure 6. Chl *a* in the sample was removed entirely after 4 h of the experiment when quencher was not added. After the addition of benzoquinone (BQ), photocatalytic algal removal was significantly inhibited, and the removal rate of Chl *a* was less than 40% after 6 h, indicating that $O_2^{\bullet-}$ was the main active species in the reaction process. This may be because BQ converted $O_2^{\bullet-}$ into $O_2$ in the reaction process, preventing the destruction of algal cells by $O_2^{\bullet-}$. After sulfur monoxide (SO) was added, the trend of photocatalytic algal removal was consistent with that of the experimental group, with a slight decrease in the rate, but no obvious inhibitory effect. This indicated that $\bullet$OH plays a role in the reaction, but it is not the main active material. After the addition of isopropyl alcohol (IPA), the inhibitory effect of IPA on the removal of Chl *a* was almost negligible, and all of the Chl *a* was also removed at 4 h. This illustrates that $O_2^{\bullet-}$ is the predominant active substance responsible for the photocatalytic process.

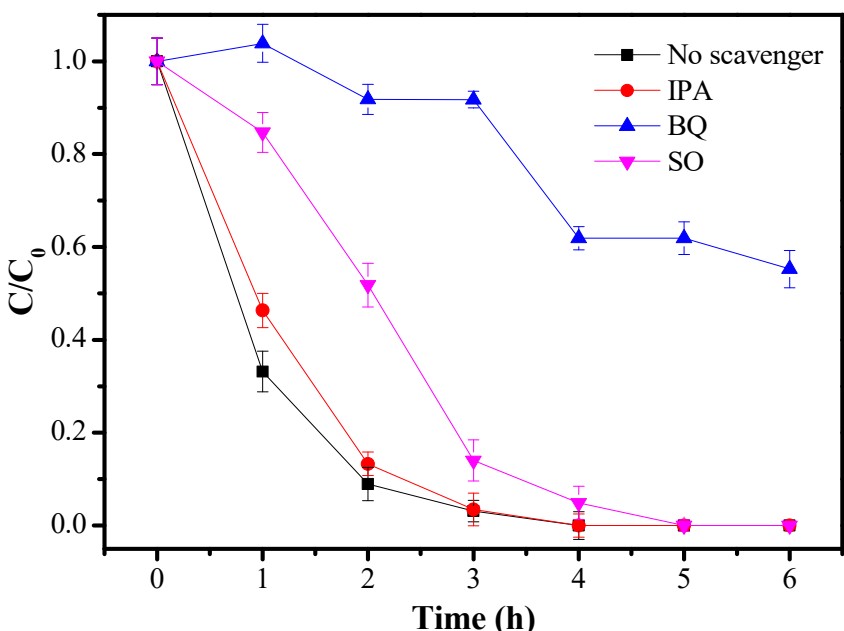

**Figure 6.** Effect of quenchers on photocatalytic activity of Ag/AgCl@ZIF-8 coating.

According to the quenching experiment results, a possible mechanism of Ag/AgCl@ZIF-8 coating on algae is proposed according to Figure 7. ZIF-8 cannot be inspired under sunlight for its wide energy band. However, it is possible for Ag/AgCl by the condition that the incident photon energy is greater than the bandgap value of Ag/AgCl. After that, the electrons transfer from the valence band (VB) to the conduction band (CB) forming $h^+$ in the VB. Therefore, $h^+$ reacts with $H_2O$ to form $\bullet$OH, which inactivates the algal cells. It is speculated that $Ag^0$ is wrapped on the surface of Ag/AgCl and forms a heterojunction with ZIF-8, which forms a bridge connecting Ag/AgCl and ZIF-8, facilitating the transfer of $e^-$ to the surface of ZIF-8, achieving effective separation of electron-hole pairs, and reducing

the recombination to some extent [35]. Only when the CB position of materials is more negative than $O_2/O_2^{\bullet-}$, can $O_2$ be reduced to $O_2^{\bullet-}$, and the more negative the position is, the easier it is to reduce $O_2$. The CB potential of Ag/AgCl (−0.09 eV vs. NHE) is close to that of $O_2/O_2^{\bullet-}$ (−0.046 eV vs. NHE), which is unfavorable to the formation of $O_2^{\bullet-}$, while the CB potential of ZIF-8 (−0.86 eV vs. NHE) is more negative than $O_2/O_2^{\bullet-}$ [41]. Therefore, $e^-$ reacts with $O_2$ adsorbed on the surface of ZIF-8 to form $O_2^{\bullet-}$, which acts on the cell wall and cell membrane and thus leads to cell rupture.

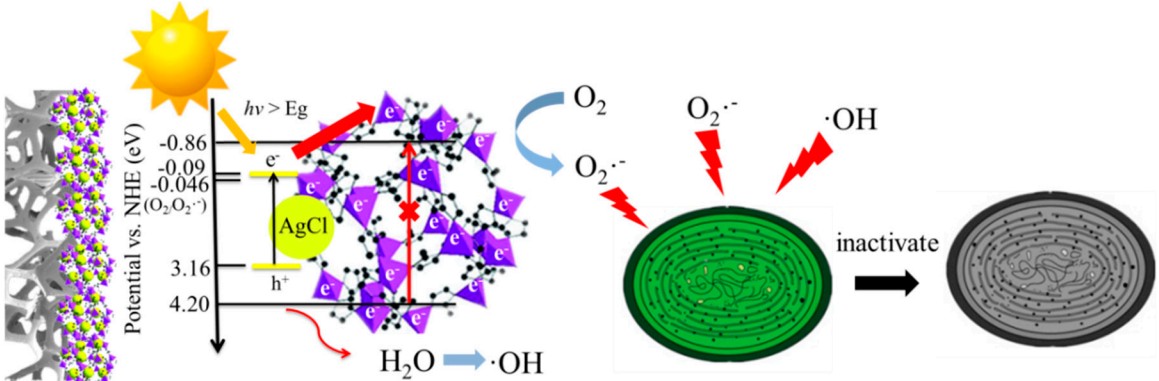

**Figure 7.** Possible mechanism of the Ag/AgCl@ZIF-8 coating on the photocatalytic removal of algae.

## 3. Materials and Methods

### 3.1. Synthesis of Ag/AgCl@ZIF-8 Photocatalyst

Ag/AgCl@ZIF-8 was prepared following the procedures according to our previously-reported process [35]. Typically, 2.348 g of $Zn(NO_3)_2 \cdot 6H_2O$ and 5.192 g of 2-methylimidazole were dissolved in 160 mL of methanol to form solution A and solution B, respectively. Solution A was rapidly poured into solution B under magnetic stirring, and vigorously stirred at room temperature for 2 h. The resulting precipitate was harvested by centrifugation and washed three times with absolute ethanol. Finally, ZIF-8 was obtained after being kept at 343.15 K for 12 h.

0.2 g of ZIF-8 was dispersed in 14 mL of 53.7 mM $AgNO_3$ water-ethanol (*v/v* = 1:6) mixture under magnetic stirring for 3 h. Then the solution was added dropwise into 98 mL of 10.48 mM NaCl water-ethanol (*v/v* = 1:6) mixture within 20 min and stirred at room temperature for 10 h. The solution was changed from white to light blue. In order to get Ag/AgCl@ZIF-8, the resultant products were washed three times with deionized water to remove excess sodium chloride followed by drying at 343.15 K for 12 h.

### 3.2. Preparation of Ag/AgCl@ZIF-8 Coating

The Ag/AgCl@ZIF-8 coating was prepared by the dip-coating method (Figure S1). Two pre-washed sponges (4 cm × 4 cm) were immersed in 50 mL of 3.0 g/L sodium dodecyl benzene sulfonate (SDS) solution, oscillated on a shaker for 30 min, then washed to remove excess SDS and dried at drying oven to obtain the SDS-modified sponges. Then, two SDS-modified sponges were immersed in 50 mL of 2.5 g/L Ag/AgCl@ZIF-8 aqueous solution, oscillating on a shaker for 2 h. Finally, the Ag/AgCl@ZIF-8 coating was given after being dried at 363.15 K for 12 h while turning over the sponge every few minutes until the water on the surface evaporates completely to prevent the catalyst precipitating.

### 3.3. Characterization

The X-ray diffraction (XRD) patterns were identified by X-ray diffractometry (Rigaku, Ultima IV, Tokyo, Japan) using Cu Kα radiation (λ = 0.15406 Å, 40 mA, 40 kV) at a scanning speed of 20°/min, in the 2θ range of 5–80°. UV-visible (UV-vis) diffuse reflection spectra were obtained using a UV-vis spectrophotometer (Perkin-Elmer, Lambda950, Waltham, MA, USA) in the range of 300–800 nm with

$BaSO_4$ as a reflectance standard. The size and morphology of the samples were observed by scanning electron microscope (SEM) (Hitachi, S-4800, Tokyo, Japan) with an acceleration voltage of 5 kV and an acceleration current of 7 μA.

### 3.4. Algae Density and Diversity Index

The diluted algae solution was fixed with a certain amount of Lugol's iodine solution, and a 0.1 mL sample was taken under the optical microscope for plankton count analysis. The intelligent identification system software (WSeen, AlgaeC, Hangzhou, China) was used to identify species of algae and calculate the algae density, phytoplankton biomass, Shannon-Werner index, richness index and uniformity index automatically.

### 3.5. Natural Water Samples

The natural water samples were taken from the landscape lake (Figure S2) in Fuzhou University Zhicheng College, located in Gulou District, Fuzhou (119°16′12″E, 26°04′38″N). The area of the lake is about 7,536 $m^2$, and the depth is about 2.0 m. The pH, turbidity, and temperature of the lake water were measured on the spot. The water samples were taken back for the basic water quality indices determination (Table S1) and algae species identification. The lake has a large amount of suspended matter and high turbidity, belonging to a heavily eutrophic water body.

A total of 6 phyla and 26 genera of phytoplankton were found in the natural lake water (Figure S3), and there were 10 genera and 25 species which belonged to *Chlorophyta*, 7 genera and 12 species belonged to *Bacillariophyta*, 4 genera with 8 species belonged to *Cyanophyta*, 2 genera and 6 species belonged to *Euglenophyta*, 1 genus and 3 species belonged to *Crytophyta* and 2 genera along with 2 species belonged to *Chrysophyta*, accounting for 44.6%, 21.4%, 14.3%, 10.7%, 5.36% and 3.57% of the total species, respectively. The number of phytoplankton species in the lake is *Chlorophyta > Bacillariophyta > Cyanophyta > Euglenophyta > Cryptophyta > Chrysophyta*. The species *M. aeruginosa, Chlamydomonas reinhardtii, Pseudoanabaena mucicola* and *Nitischia palea* comprised the majority of algae species in the lake. The density of *M. aeruginosa* was $3.6 \times 10^9$ cells/L which was the highest, and much larger than the number of other algae in the water. Therefore, *M. aeruginosa* and was studied separately from other algae to analyze the effect of the coating on algae in the natural water more accurately.

### 3.6. Removal of Algae in Natural Water Samples by Coating

An Ag/AgCl@ZIF-8 coating was added in 100 mL water samples as the experimental group. The experiment was conducted under sunlight with the low-speed magnetic agitation simulating the flow turbulence of natural water bodies. The control group was conducted under the same conditions with a blank sponge. A certain amount of suspension was withdrawn every 1 h to measure the density of algae, the biomass and species of phytoplankton, and the content of chlorophyll (Chl) *a* and colored dissolved organic matter (CDOM).

#### 3.6.1. Chlorophyll *a*

The content of chlorophyll *a* was measured, referring to China EPA standard methods [42]. 8 mL of the sample was taken at 0, 1, 2, 3, 4, 5 and 6 h after the experiment; then a suction filtration was used to attach algae cells onto cellulose acetate membranes. These filter membranes were put into the centrifuge tubes, then frozen for 1 h at 253.15 K and thawed 1 h at room temperature. This process was repeated three times. Centrifuge tubes were vibrated in a vortex mixer for 20 s after adding 90% acetone and being refrigerated at 277.15 K for 4 h in darkness. After that, these samples were vibrated in the vortex for 20 s and centrifuged at 3500/min for 15 min, then the UV-visible spectrophotometer (Shimadzu, UV-Vis2450, Kyoto, Japan) was employed to measure OD630, OD647, OD664 and OD750 of the supernatant. The Chl *a* content was calculated following Equation (1).

$$\text{Chlorophyll} a (\text{mg/L}) = \frac{[11.85(\text{OD}_{664} - \text{OD}_{750}) - 1.54(\text{OD}_{647} - \text{OD}_{750}) - 0.08(\text{OD}_{630} - \text{OD}_{750})]V_1}{V_2 L} \qquad (1)$$

### 3.6.2. Colored Dissolved Organic Matter

The determination of CDOM in samples was analyzed by a three-dimensional fluorescence spectrometer (Edinburgh, FS5, Edinburgh, UK). Xenon lamp was used as the excitation light source. The excitation wavelength was 200–550 nm, the emission wavelength was 200–600 nm, the slit width was 5 nm, and the step distance was 5 nm. Deionized water was used as the blank correction detection result.

### 3.6.3. Reactive Oxygen Species

The reactive oxygen species (ROSs) produced in the process of photocatalytic experiments can be characterized by free radical quenching experiments. The experiment was based on the fact that quenching agents can reduce the photocatalytic efficiency by capturing free radicals or competing with free radicals [43]. It has been widely found that $\bullet OH$, $O_2^{\bullet-}$ and $h^+$ are the main ROSs in a photocatalytic reaction. Therefore, 1 mM isopropyl alcohol (IPA), benzoquinone (BQ) and sulfur monoxide (SO) were added into 100 mL samples as the quenching agent of $\bullet OH$, $O_2^{\bullet-}$ and $h^+$, respectively [44]. Then the content of Chl *a* was measured every hour to indicate the photocatalytic activity of the coating.

## 4. Conclusions

Ag/AgCl@ZIF-8 can be loaded effectively and uniformly onto sponges treated by SDS with a dip-coating method. Ag/AgCl@ZIF-8 coating can effectively reduce the density and biomass of phytoplankton, change the community structure of phytoplankton, and decrease the proportion of *Cyanobacteria* in the algal blooms water. In this study, the density and biomass of phytoplankton in water samples except *M. aeruginosa* were reduced by 78.8% and 82.7% in 6 h, respectively. The density of *M. aeruginosa* decreased to 7.4%, and 99.9% of Chl *a* was removed. All of the Shannon-Wiener index, richness index and uniformity index show an attenuation trend, and the diversity of phytoplankton in water was reduced. In addition, the coating can reduce the content of CDOM in water, degrading algal metabolites and reducing the possibility of a recurrence of algal blooms. After four recycling experiments, the removal rate of Chl *a* by coating can reach 94%, indicating that the coating had a good recycling performance. In the reaction, the contribution of ROSs is $O_2^{\bullet-} > h^+ > \bullet OH$. A series of injuries on physiological metabolism occurred with the accumulation of ROSs in algal cells. The boundaries between cells and the outline of cells became blurred, and the cell structure was dissolved. This study has demonstrated Ag/AgCl@ZIF-8 coating as an effective solution to cyanobacterial blooms and provided a novel water remediation method to remove algae and pollutants from water.

**Supplementary Materials:** The following are available online at http://www.mdpi.com/2073-4344/9/8/698/s1, Chemicals and reagents, Figure S1: Dip-coating preparation process, Figure S2: (a) Photographs of lakes and (b) sampling points for actual water sampling, Figure S3: Species composition of phytoplankton in natural landscape lake, Figure S4: XRD patterns of Ag/AgCl@ZIF-8 powder and coating, Figure S5: Diversity index of algae during photocatalytic algal removal process: (a) Shannon-Werner index, (b) richness index and (c) uniformity index, Figure S6: Reusability of Ag/AgCl@ZIF-8 coating for photodegradation of chlorophyll a, Table S1: Basic water quality index, Table S2: Comparison with other photocatalyst reported in the literature.

**Author Contributions:** G.F., S.W. and Z.Z. conceptualized and supervised. Z.C. and X.Z. performed experimental work. All the authors contributed in writing, editing, and analysis.

**Funding:** This research was funded by the National Natural Science Foundation of China (No. 51778146 and No. 51778082), the Outstanding Youth Fund of Fujian Province in China (No. 2018J06013), the Technical Research Project of Shenzhen Municipal Science and Technology Innovation Commission in China (No. 20170422), and the Open Project Program of National Engineering Research Center for Environmental Photocatalysis (No. NERCEP-201901).

**Acknowledgments:** The authors would like to thank the reviewers and editors for their valuable remarks and comments that greatly improved the quality of the paper.

**Conflicts of Interest:** There are no conflicts to declare.

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
