# Peer review of "Photocatalytic Removal of Harmful Algae in Natural Waters by Ag/AgCl@ZIF-8 Coating under Sunlight"

_catalysts, doi:10.3390/catal9080698_

Round 1
Reviewer 1 Report
Dear Authors:
The article about Ag/AgCl@ZIF-8 as a photocatalyst is interesting and shows its functionality versus harmful algae in water. Nevertheless, under my knowledge, some changes should be done for improving the manuscript.
Introduction section, page 1, line 40. It is mentioned that ”Photocatalysis is the acceleration of a chemical reaction stimulated by light with the aid of photocatalyst, which is usually a semiconductor”. In this sense, it should be added that the combination of semiconductors and the way of combined them is also important for the catalysts response. Moreover, the dispersion grade of semiconductors and its contact with the catalyst or support presents influence on reaction efficiency and on the band gap. In this sense, one phrase may be added and a reference as their dispersion presents influence:
Reinosa, J. J.; Álvarez-Docio, C. M.; Zapata-Ramírez, V; Fernández J. F.; Hierarchical nano ZnO-micro TiO2 composites: High UV protection yield lowering photodegradation in sunscreens. Ceramics International, 44(3) (2018) 2827-2834.
In figure 1, a micrograph with higher magnification should be showed in order to observe the covering of the sponge by the catalyst.
In line 290, the units of lambda should me added.
In figure 7, OH radicals appear but they are not described in the manuscript. Please, modify the figure or add an explanation in the manuscript.
Best regards
Author Response
Dear editor and reviewers,
Thanks a lot for your attentions paid to this work. We do appreciate the professional and illuminating comments provided by the reviewers. These advices and suggestions are valuable for both this work and our subsequent research. We have tried our best to revise the manuscript (Manuscript ID catalysts-570175) based on the comments. We hope the revision can be satisfactory. All changes made in the revised manuscript are highlighted in red font. The illustration of the revisions is shown point by point as follows.
Finally, thank you very much for your comments and suggestions again.
Best regards,
Gongduan Fan
2019-8-10
Response to Reviewer: 1
Introduction section, page 1, line 40. It is mentioned that “Photocatalysis is the acceleration of a chemical reaction stimulated by light with the aid of photocatalyst, which is usually a semiconductor”. In this sense, it should be added that the combination of semiconductors and the way of combined them is also important for the catalysts response. Moreover, the dispersion grade of semiconductors and its contact with the catalyst or support presents influence on reaction efficiency and on the band gap. In this sense, one phrase may be added and a reference as their dispersion presents influence:
Reinosa, J. J.; Álvarez-Docio, C. M.; Zapata-Ramírez, V; Fernández J. F.; Hierarchical nano ZnO-micro TiO2 composites: High UV protection yield lowering photodegradation in sunscreens. Ceramics International, 44(3) (2018) 2827-2834.
Response: Thanks for the comment. The authors have read this paper and added some phrases to describe the influence of dispersion of photocatalysts. The change could be found in Line 44-46, Page 1 and ref 12.
In figure 1, a micrograph with higher magnification should be showed in order to observe the covering of the sponge by the catalyst.
Response: Thanks for the valuable comment. The authors have added two SEM images of the blank sponges and Ag/AgCl@ZIF-8 coating with higher magnification. The covering of the sponge by the catalyst can be observed more clearly from Figure 1 (e) and (f).
In line 290, the units of lambda should me added.
Response: Thanks for your carefulness. Sorry for our carelessness. The units of lambda have been added and can be found in Line 226, Page 7.
In figure 7, OH radicals appear but they are not described in the manuscript. Please, modify the figure or add an explanation in the manuscript.
Response: Thanks for the valuable comment. In quenching experiment, the trend of photocatalytic algal removal was consistent with that of the experimental group after SO was added, with a slight decrease in the rate, but no obvious inhibitory effect. This indicated that •OH plays a role in the reaction, but it is not the main active material. The authors have added the explanation of the generation and effect of •OH in photocatalytic reactions. The changes can be found in Line 257-258 and Line 268-269,Page 8.

Reviewer 2 Report
The manuscript written by Fan et al. concerns photocatalytic properties of Ag/AgCl/ZIF-8 coating in the removal of algae. The results are significant. The manuscript can be published after the following minor revisions:
1. The selection of Ag/AgCl as a modifier for ZIF-8 should be more explained. It is clear that it improves light absorption properties of ZIF-8 but why this modifier was used and not other.
2. Did Authors consider other types of carriers than sponges? The role of carrier (sponge) should be also more clarified in the manuscript.
3. What was the reason for the treatment of sponges by SDS solution? There is no information about it in the text.
Author Response
Response to reviewer's comments
Dear editor and reviewers,
Thanks a lot for your attentions paid to this work. We do appreciate the professional and illuminating comments provided by the reviewers. These advices and suggestions are valuable for both this work and our subsequent research. We have tried our best to revise the manuscript (Manuscript ID catalysts-570175) based on the comments. We hope the revision can be satisfactory. All changes made in the revised manuscript are highlighted in red font. The illustration of the revisions is shown point by point as follows.
Finally, thank you very much for your comments and suggestions again.
Best regards,
Gongduan Fan
2019-8-10
Response to Reviewer: 2
The selection of Ag/AgCl as a modifier for ZIF-8 should be more explained. It is clear that it improves light absorption properties of ZIF-8 but why this modifier was used and not other.
Response: Thanks for your comment. Due to the electron trapping ability and localized surface plasmon resonance (LSPR) effect of silver nanoparticles[1], Ag and Ag compounds are often used as a modified material for the synthesis of catalysts or heterostructures[2]. In our previous study[3], MOFs modified by Ag/AgCl have a good degradation effect on pharmaceutical and personal care products (PPCPs). Therefore, Ag/AgCl was selected as the modifier for ZIF-8 to study its effect and mechanism of algae removal in this paper. To address this comment, the authors have explained the reason why they choose Ag/AgCl as a modifier for ZIF-8. The change can be found in Line 75-80,Page 2.
Did Authors consider other types of carriers than sponges? The role of carrier (sponge) should be also more clarified in the manuscript.
Response: Thanks for the comment. In order to select a suitable carrier, the authors have tried variety of load materials (such as small glass bottles, foamed nickel, cork and sponges) and loading methods (in site growth, binder, dip-binder and dip-coating method). The result show that it can more effectively and uniformly load Ag/AgCl@ZIF-8 on sponges with dip-coating method than other carriers and methods. As a common load material, sponge is easy to promote in practical applications for its advantages of low cost and easy access. Additionally, the sponge is a porous material that can float on the surface of the water and adsorbs a large amount of M. aeruginosa cells, which is beneficial for the material to absorb sunlight and contact with algae on the surface of the water body. Floating and coating material can not only improve the efficiency of photocatalytic algae removal but also is beneficial to the recovery and reuse to avoid secondary pollution. To address this comment, the authors have clarified the role of carrier (sponge). The change can be found in Line 83-89,Page 2.
What was the reason for the treatment of sponges by SDS solution? There is no information about it in the text.
Response: Thanks for the comment. According to our previous studies[3], the zeta potential of Ag/AgCl @ZIF-8 exhibits a positive surface charge in a neutral environment. SDS is a commonly used anionic surfactant that can attract Ag/AgCl@ZIF-8 to the sponge treated with SDS by electrostatic effect. Meanwhile, 2-methylimidazole of ZIF-8 contains an imidazole ring, which can also be attracted to the surface of the sponge by π-π interaction with the benzene ring of SDS[4]. To address this comment, the authors have explained the function of SDS. The change can be found in Line 122-127,Page 4.
References
Wang, W.; Li, G.; Xia, D.; An, T.; Zhao, H.; Wong, P.K. Photocatalytic nanomaterials for solar-driven bacterial inactivation: recent progress and challenges. Environmental Science: Nano 2017, 4, 782-799. Chen, F.; Yang, Q.; Wang, Y.; Zhao, J.; Wang, D.; Li, X.; Guo, Z.; Wang, H.; Deng, Y.; Niu, C., et al. Novel ternary heterojunction photcocatalyst of Ag nanoparticles and g-C3N4 nanosheets co-modified BiVO4 for wider spectrum visible-light photocatalytic degradation of refractory pollutant. Applied Catalysis B: Environmental 2017, 205, 133-147. Fan, G.D.; Zheng, X.M.; Luo, J.; Peng, H.P.; Lin, H.; Bao, M.C.; Hong, L.; Zhou, J.J. Rapid synthesis of Ag/AgCl@ZIF-8 as a highly efficient photocatalyst for degradation of acetaminophen under visible light. Chem. Eng. J. 2018, 351, 782-790. Wen, M.; Li, G.; Liu, H.; Chen, J.; An, T.; Yamashita, H. Metal–organic framework-based nanomaterials for adsorption and photocatalytic degradation of gaseous pollutants: recent progress and challenges. Environmental Science: Nano 2019, 6, 1006-1025.

Reviewer 3 Report
The manuscript entitled "Photocatalytic removal of harmful algae in natural
3 waters by Ag/AgCl@ZIF-8 coating under sunlight" describes the catalytic efficiency of Ag/AgCl@ZIF-8 coating to remove the natural bloom problem under sunlight.
The manuscript is interesting and can be accepted after a major revision as noted.
Scale bar is needed for Fig. 1 a, b. The main issue is that the degradation time is huge. May be some positive direction for future work to make the catalysis faster should be discussed. For instance, the incorporation of some metal-oxide nanopartcile/quantumdot might enhance the photocatalytic efficiency when mixed with the system. The microstructure of sponge reveals its macroporosity. This is an important factor for the photocatalysis. These pore may helps in transportation of reactant/product during the reaction. Also, porous surface allows immobilization of more catalysts for more surface reaction sites. A comparison table of the reported photocatalysis efficiency and others reported in the literature should be provided for the photocatalysts. This could provide a clear picture for the photocatalytic efficiency of the photocatalysts for pollutant removal.Author Response
Dear editor and reviewers,
Thanks a lot for your attentions paid to this work. We do appreciate the professional and illuminating comments provided by the reviewers. These advices and suggestions are valuable for both this work and our subsequent research. We have tried our best to revise the manuscript (Manuscript ID catalysts-570175) based on the comments. We hope the revision can be satisfactory. All changes made in the revised manuscript are highlighted in red font. The illustration of the revisions is shown point by point as follows.
Finally, thank you very much for your comments and suggestions again.
Best regards,
Gongduan Fan
2019-8-10
Response to Reviewer: 3
Scale bar is needed for Fig. 1 a, b.
Response: Thanks for the comment. Sorry for our careless. The size of a sponge is 4 cm×4 cm. The authors have added the scale bar in Figure 1. (a) and (b). And the explanation of the sponge’s size has been added in Section 3.2, Line 304, Page 9.
The main issue is that the degradation time is huge. May be some positive direction for future work to make the catalysis faster should be discussed. For instance, the incorporation of some metal-oxide nanopartcile/quantumdot might enhance the photocatalytic efficiency when mixed with the system.
Response: Thanks for the comment. It is a very worthy research project to modify the photocatalyst by nanoparticle or quantumdot. The authors strongly agree these positive directions for future work can make the catalysis faster. The authors hope to enhance the effectiveness of the photocatalyst through different methods in future work. The Ag/AgCl@ZIF-8 coating have performed significant effect on algae removal. The reasons for the long degradation time are as follows: (1) The experiments were carried out in natural water samples with more complex substrate background and various species of algae, which will affect the photocatalytic removal of M. aeruginosa. (2) Photocatalytic experiments were carried out with natural water samples under sunlight, so the light intensity may varies during the process. (3) The initial algal density of M. aeruginosa was 3.6×106 cells/mL in the natural water sample, which is more than that (general is 2.7×106 cells/mL) in other paper reported. There were other phytoplankton (the initial algal density was 55.9×106 cells/L) in the samples which would consume a part ofthe active species produced by the photocatalyst. (4) The dose of photocatalyst loaded on every sponge is about 0.06g amount to the equivalent of about 0.6g/L, which is almost 1/3 or 1/7 of the dose reported in other literature. And a comparison with other photocatalyst reported in the literature has been shown in Table S2.
Table S2 Comparison with other photocatalyst reported in the literature
Catalyst |
Catalyst Dose (g/L) |
Microbial level (cell/mL) |
Light source |
Reaction time (h) |
Final inactivation rate |
Ref |
g-C3N4/Al2O3/EP |
2 |
2.7 × 106 |
500 W Xenon lamp with a UV filter |
6 |
74.40% |
[1] |
g-C3N4/TiO2/Al2O3/EP |
2 |
2.7 × 106 |
6 |
88.10% |
[2] |
|
TiOX-550 (X=N and P) |
4 |
2.7 × 106 |
6 |
81.5% |
[3] |
|
CeOx/TiO2-yFy |
4 |
2.7 × 106 |
4 |
100% |
[4] |
|
NPTiO2 |
4 |
2.7 × 106 |
6 |
92.60% |
[5] |
|
Bi2Mo3O12 |
1 |
/ |
4 |
84% (Chl a) |
[6] |
|
NTiO2 |
2 |
5 × 106 |
LED lamp |
14 |
100% |
[7] |
Ag/AgCl@ZIF-8 coating |
0.625 |
3.6×106 |
sunlight |
6 |
92.60% |
|
99.9% (Chl a) |
|
3. The microstructure of sponge reveals its macroporosity. This is an important factor for the photocatalysis. These pore may helps in transportation of reactant/product during the reaction. Also, porous surface allows immobilization of more catalysts for more surface reaction sites.
Response: Thanks for the valuable comment. The authors strongly agree that porous surface of sponge paly a important role in the photocatalysis process. It can make the coating float on the surface of the water and adsorbs large number of algae cells, which is beneficial for the material to absorb sunlight and contact with algae on the surface of the water body. These pores may help in transportation of reactant/product during the reaction. Also, porous surface allows immobilization of more catalysts for more surface reaction sites. The authors have added the explaination about the role of sponge’s macroporosity. The change could be found in Line 86-89, Page.2 and Line 121-123, Page.4.
A comparison table of the reported photocatalysis efficiency and others reported in the literature should be provided for the photocatalysts. This could provide a clear picture for the photocatalytic efficiency of the photocatalysts for pollutant removal.
Response: Thanks for the valuable comment. The authors hcve compared the performance of other photocatalysts and Ag/AgCl@ZIF-8 coating in algae removal. The coating performed significant effect on algae removal with the lowest dose and high removal rate. And a comparison with other photocatalyst reported in the literature has been shown in Table S2. And the description of the comparison has been added in Line 184-187, Page 6.
Table S2 Comparison with other photocatalyst reported in the literature
Catalyst |
Catalyst Dose (g/L) |
Microbial level (cell/mL) |
Light source |
Reaction time (h) |
Final inactivation rate |
Ref |
g-C3N4/Al2O3/EP |
2 |
2.7 × 106 |
500 W Xenon lamp with a UV filter |
6 |
74.40% |
[1] |
g-C3N4/TiO2/Al2O3/EP |
2 |
2.7 × 106 |
6 |
88.10% |
[2] |
|
TiOX-550 (X=N and P) |
4 |
2.7 × 106 |
6 |
81.5% |
[3] |
|
CeOx/TiO2-yFy |
4 |
2.7 × 106 |
4 |
100% |
[4] |
|
NPTiO2 |
4 |
2.7 × 106 |
6 |
92.60% |
[5] |
|
Bi2Mo3O12 |
1 |
/ |
4 |
84% (Chl a) |
[6] |
|
NTiO2 |
2 |
5 × 106 |
LED lamp |
14 |
100% |
[7] |
Ag/AgCl@ZIF-8 coating |
0.625 |
3.6×106 |
sunlight |
6 |
92.60% |
|
99.9%(Chl a) |
|
References
Song, J.; Wang, X.; Ma, J.; Wang, X.; Wang, J.; Zhao, J. Visible-light-driven in situ inactivation of Microcystis aeruginosa with the use of floating g-C3N4 heterojunction photocatalyst: Performance, mechanisms and implications. Appl Catal B-Environ 2018, 226, 83-92. Song, J.; Wang, X.; Ma, J.; Wang, X.; Wang, J.; Xia, S.; Zhao, J. Removal of Microcystis aeruginosa and Microcystin-LR using a graphitic-C3N4/TiO2 floating photocatalyst under visible light irradiation. Chem. Eng. J. 2018, 348, 380-388. Wang, X.; Wang, X.; Song, J.; Li, Y.; Wang, Z.; Gao, Y. A highly efficient TiOX (X = N and P) photocatalyst for inactivation of Microcystis aeruginosa under visible light irradiation. Sep. Purif. Technol. 2019, 222, 99-108. Wang, X.; Song, J.; Su, C.; Wang, Z.; Wang, X. CeOx/TiO2-yFy nanocomposite: An efficient electron and oxygen tuning mechanism for photocatalytic inactivation of water-bloom algae. Ceram. Int. 2018, 44, 19151-19159. Wang, X.; Song, J.; Zhao, J.; Wang, Z.; Wang, X. In-situ active formation of carbides coated with NPTiO2 nanoparticles for efficient adsorption-photocatalytic inactivation of harmful algae in eutrophic water. Chemosphere 2019, 228, 351-359. Li, P.; Wang, L.; Liu, H.; Li, R.; Xue, M.; Zhu, G. Facile sol-gel foaming synthesized nano foam Bi2Mo3O12 as novel photocatalysts for Microcystis aeruginosa treatment. Mater. Res. Bull. 2018, 107, 8-13. Jin, Y.; Zhang, S.; Xu, H.; Ma, C.; Sun, J.; Li, H.; Pei, H. Application of N-TiO2 for visible-light photocatalytic degradation of Cylindrospermopsis raciborskii - More difficult than that for photodegradation of Microcystis aeruginosa ? Environ Pollut 2019, 245, 642-650.

Reviewer 4 Report
The authors have described in detail and with very well described experimental method, a catalyst using a relatively benign material, Ag/AgCl@ZIF-8 material. The paper is very well written, and has proper control experiments, and has shown the the material in question can destroy harmful bacteria based on cyanide in natural waters with sunlight. This research can become increasingly important as climate adaptations occur in the coming generations.
The conclusion can be more inclusive of some of the results obtained.
As a future or follow up- study,it would be interesting to know how temperature affects the performance (since intraday temperatures can vary by a few degrees or several dozens), and if this material can be used effectively for other micro organisms such a E. Coli, for example.
Author Response
Response to reviewer's comments
Dear editor and reviewers,
Thanks a lot for your attentions paid to this work. We do appreciate the professional and illuminating comments provided by the reviewers. These advices and suggestions are valuable for both this work and our subsequent research. We have tried our best to revise the manuscript (Manuscript ID catalysts-570175) based on the comments. We hope the revision can be satisfactory. All changes made in the revised manuscript are highlighted in red font. The illustration of the revisions is shown point by point as follows.
Finally, thank you very much for your comments and suggestions again.
Best regards,
Gongduan Fan
2019-8-10
Response to Reviewer: 3
The conclusion can be more inclusive of some of the results obtained.
Response: Thanks for the comment. The authors have added the description about the results of characterization, diversity of phytoplankton and morphological changes in algal cells in conclusion section. The change could be found in Line 367-368, Line 372-375,and Line 379-380 page 11
As a future or follow up- study, it would be interesting to know how temperature affects the performance (since intraday temperatures can vary by a few degrees or several dozens), and if this material can be used effectively for other microorganisms such as E. coli, for example.
Response: Thanks for the valuable comment. The author agrees that the influence of temperature on the experimental results is very important, and they will consider the influence of temperature in the subsequent research. At the same time, the author is very grateful for the reviewer's suggestion to apply the coating to other microorganisms, which is a very worthy research project. It was reported that silver compounds and MOFs show excellent bactericidal performance. Li. et al.[1] have found that ZIF-8 exhibits almost complete inactivation of Escherichia coli (E. coli) (>99.9999% inactivation efficiency) in saline within 2 h of simulated solar irradiation. Cheikhrouhou. et al.[2] have found that Ag/AgCl exhibited strong antibacterial activity against both with complete inactivation 1.5 × 107 cfu mL−1 of E. coli within 15 min. Therefore, the authors believe that Ag/AgCl@ZIF-8 could be used effectively for other microorganisms (such as E. coli) and they hope to conduct related research in the future.
References
Li, P.; Li, J.; Feng, X.; Li, J.; Hao, Y.; Zhang, J.; Wang, H.; Yin, A.; Zhou, J.; Ma, X., et al. Metal-organic frameworks with photocatalytic bactericidal activity for integrated air cleaning. Nat. Commun. 2019, 10, 2177. Cheikhrouhou, W.; Kannous, L.; Ferraria, A.M.; Botelho do Rego, A.M.; Kamoun, A.; Rei Vilar, M.; Boufi, S. AgCl/Ag functionalized cotton fabric: An effective plasmonic hybrid material for water disinfection under sunlight. SoEn 2019, 183, 653-664.

Round 2
Reviewer 3 Report
The authors have addressed all concerns and the manuscript can now be accepted.